



# Glaciogenic seeding-induced hole-punch clouds and their sensitivity to the clouds' background state

Nadja Omanovic[1], Debora Bötticher[1], Christopher Fuchs[1], and Ulrike Lohmann[1]

[1]Institute for Atmospheric and Climate science, ETH Zurich, Zurich, Switzerland

**Correspondence:** Nadja Omanovic (nadja.omanovic@env.ethz.ch) and Ulrike Lohmann (ulrike.lohmann@env.ethz.ch)

**Abstract.** Hole-punch clouds are a visual representation of ice crystal formation and growth as well as their interactions with the liquid phase via the Wegener–Bergeron–Findeisen process. Their appearance usually is associated with an aircraft passing through a liquid cloud layer. However, they can also appear upon glaciogenic seeding of a supercooled low-level stratus cloud, as we showcase in this study. The observations of a hole-punch cloud prompted an investigation into the sensitivity of these
clouds to the clouds' background state. We employ high-resolution large-eddy simulations with the weather model ICON and simulate one seeding experiment with different initial liquid water paths. The ensemble of nine simulations helps us to quantify how the properties of a hole-punch cloud, i.e., strong reductions in the liquid water contents, depends on the cloud liquid water paths. Moreover, we perturb the turbulent mixing length, i.e., the Smagorinsky constant, to evaluate the impact of the intensity of mixing on hole-punch clouds. Generally, larger liquid water paths lead to a delay in the strongest reductions of liquid water
content through the Wegener–Bergeron–Findeisen process. Interestingly, the turbulent mixing length does not appear to have a significant impact on the hole-punch cloud properties. These findings improve our understanding of the Wegener–Bergeron–Findeisen process and of the effectiveness of glaciogenic cloud seeding.

## 1 Introduction

Hole-punch clouds are typically associated with circular or linear voids after an aircraft penetrates through a supercooled
liquid cloud layer (Heymsfield et al., 2010). Prior to the formation of the hole, this cloud is void of any ice because of missing aerosol particles, i.e., ice-nucleating particles (INPs), that help form ice crystals. However, the passage of an aircraft results in an inadvertent seeding effect of the cloud. The expansion of air behind an aircraft can lead to a cooling by 20 °C to 30 °C, often enough to kick-start homogeneous ice nucleation, bypassing the need for any INPs (Heymsfield et al., 2011). Upon the formation of ice crystals in a supercooled liquid cloud, the subsequent evolution of a hole-punch cloud depends on the
interactions between cloud droplets and ice crystals. One such interaction is the Wegener–Bergeron–Findeisen (WBF) process (Wegener, 1911; Bergeron, 1935; Findeisen, 1938) and it is known to be crucial for the glaciation of a cloud as well as for continental precipitation formation. It takes place because of different saturation water vapor pressures over water and ice. At water-subsaturated conditions cloud droplets will evaporate, while ice crystals continue to grow as long as they experience ice supersaturated conditions. Moreover, the evaporating cloud droplets serve as a reservoir of water vapor, which can be taken
up by the ice crystals. Eventually, ice crystals will start to sediment and form fallstreaks below the cloud. This terminates



the extent of the hole, because no more interactions with the cloud droplets are possible. Exactly this process-chain was also observed downstream of industrial areas, which emit anthropogenic aerosols (Toll et al., 2024). They observed a reduction in shortwave reflection as well as in cloud cover and cloud optical thickness.

The importance of the conversion of liquid to ice through the WBF process was early-on recognized in the weather modification community. Schaefer (1946) conducted first experiments of ice crystals in supercooled clouds and Vonnegut (1947) identified the importance of silver iodide acting as an INP to form ice crystals already at $-5\,°C$. At these temperatures, very few natural INPs exist, such that silver iodide is typically used in targeted cloud seeding operations. Across the globe cloud seeding programs emerged with focus on scientific understanding as well as operational usage for precipitation enhancement (Haupt et al., 2018). Within the scientific community, a project called "Seeded and Natural Orographic Wintertime Clouds: The Idaho Experiment" demonstrated the microphysical process-chain upon seeding orographic clouds using remote sensing instruments (French et al., 2018). They also highlighted fundamental knowledge gaps in our understanding of mixed-phase cloud processes. Building on their findings, the CLOUDLAB project aims to address these knowledge gaps by employing the methods of weather modification, i.e., glaciogenic seeding, in a controlled and local manner (Henneberger et al., 2023). Instead of targeting orographic cloud systems, we chose a stable low-level stratus cloud to conduct our seeding experiments. This way, confounding factors, such as orographic forcing, are minimized. The focus was mainly on the quantification of ice growth rates within supercooled clouds in a natural setting (Ramelli et al., 2024; Fuchs et al., 2025). We also created a numerical modeling framework that can be used to reproduce the seeding experiments (Omanovic et al., 2024, 2025c). This framework forms the foundation for this study, such that we can assess the interplay of the liquid and ice phases across a range of cloud states. We aim to answer how the background state, in particular the liquid water content and path, of a supercooled liquid cloud defines the effectiveness of forming a hole-punch cloud by glaciogenic seeding. Additionally, we perturb the turbulent mixing length to assess the effect of more and less mixing within the model simulations. The targeted glaciogenic seeding experiments during CLOUDLAB closely follow the underlying mechanism of hole-punch clouds, as a local perturbation within a liquid cloud triggers the WBF process and changes the cloud characteristics.

## 2 Data and Methods

### 2.1 The CLOUDLAB project

Within the CLOUDLAB project, we conducted 78 glaciogenic cloud seeding experiments to infer ice crystal growth rates in supercooled low-level stratus clouds (Henneberger et al., 2023; Miller et al., 2024; Ramelli et al., 2024; Fuchs et al., 2025). This type of clouds occurs frequently in winter in Switzerland. They are often void of any ice crystals, making them the ideal candidate for cloud seeding. We employed an uncrewed aerial vehicle (UAV) upstream of the main field site to disperse the seeding material, which consists mostly of silver iodide (Miller et al., 2025). The seeding plumes containing the seeding material and the newly formed ice crystals were advected towards the main field site with the wind and were measured by an array of remote sensing and in situ instruments. This way, a thorough assessment of the cloud's state before, during, and after the seeding was achieved. A central in situ instrument within CLOUDLAB is the in-house developed holographic imager





(HOLIMO–HOLographic Imager for Microscopic Objects), which was attached to a tethered balloon system at the main field
site. From HOLIMO, we receive information on the cloud droplet and ice crystal concentrations, contents, and size distributions
in the size range between $6\,\mu$m and $2\,$mm (Ramelli et al., 2020). All particles with diameters smaller than $25\,\mu$m are classified
as cloud droplets due to the resolution limit of the instrument. The gathered data during the seeding experiments were analyzed
with $5\,$Hz and averaged to $1\,$s, while the background state (no seeding) was analyzed with $1\,$Hz.

Out of the 78 experiments, three experiments, conducted at temperatures around $-5.5\,°$C, were striking because visually a
hole could be identified while the seeding plume passed the main field site. One such example is given in Fig. 1: (a) shows
the cloud (background state) with low visibility and (b) shows the hole visible during the passage of the seeding plume. The
tethered balloon system, which carried HOLIMO, is clearly distinguishable, pointing towards the almost complete evaporation
of cloud droplets. The interplay between the ice and liquid phase is further demonstrated in Fig. 1c, which shows the measured
liquid water content (LWC, $\mathrm{g\,m^{-3}}$) and ice crystal number concentrations (ICNC, $\mathrm{cm^{-3}}$) for one of the three hole-punch
cloud seeding experiment. The liquid water path (LWP) of the background state was $\approx 25\,\mathrm{g\,m^{-2}}$. It is clearly notable that
the reductions in LWC are co-located with the presence of ice crystals. After the passage of the seeding plume, the cloud
reverts to its background state with no ice and LWC $\approx 0.15\,\mathrm{g\,m^{-3}}$. Moreover, a ceilometer, located at the main field site, also
captured the change in cloud base, as the attenuated backscatter signal moves higher up and decreases (Fig. 1d). We cannot
conclude if cloud top was lifted as the ceilometer signal could be attenuated before reaching cloud top. The uniqueness of these
experiments motivated this study to investigate how the emergence of a hole-punch cloud depends on the background cloud
state using numerical simulations.





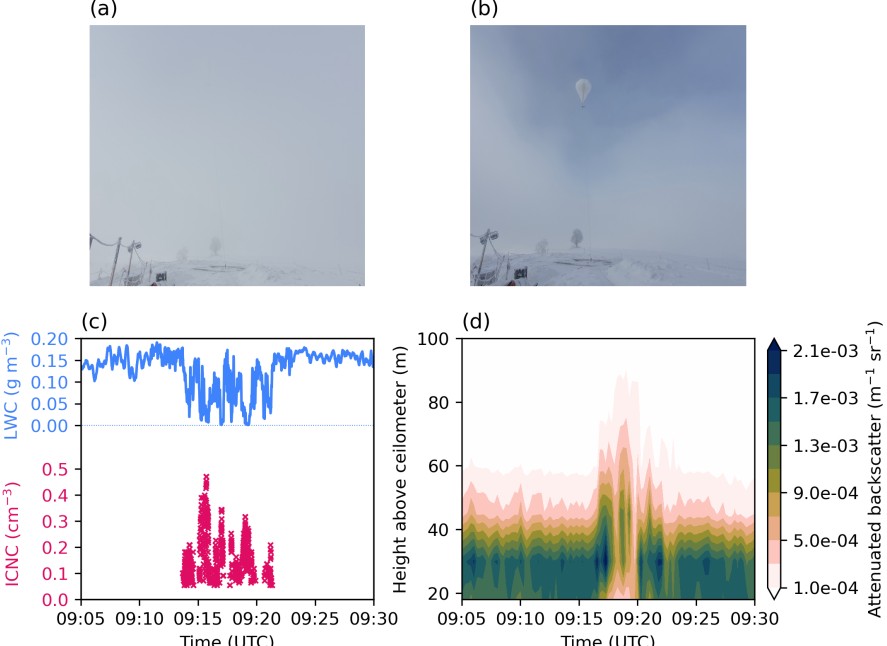

**Figure 1.** Observations of a hole-punch cloud passing over the main field site (at $920\,\mathrm{m}$ a.s.l.) at 12 January 2024. (a) and (b) show pictures taken of the cloudy background state and the hole-punch cloud. In (b), the tethered balloon system (at $1000\,\mathrm{m}$ a.s.l.) carrying the in situ instrument HOLIMO is clearly identifiable. Note that the pictures were both taken from a similar position, with the tree in the background serving as a reference point. The analyzed measurements from HOLIMO for one seeding experiment are shown in (c) in terms of liquid water content (LWC, $\mathrm{g\,m^{-3}}$, blue) and ice crystal number concentration (ICNC, $\mathrm{cm^{-3}}$, red). (d) shows the attenuated backscatter during a seeding experiment measured by a ceilometer located at the main field site.

## 2.2 Model Setup and Simulations

We already demonstrated that the seeding experiments can be successfully reproduced in numerical simulations using the weather prediction model ICON (Zängl et al., 2015) at high horizontal resolution, i.e., in large-eddy mode (Omanovic et al.,

2024). We also showed that the simulated ice crystal growth rates need to be increased by a factor 3 to achieve comparable reduction in the liquid phase as observed (Omanovic et al., 2025c). The here presented results are based on ICON simulations at a horizontal resolution of $65\,\mathrm{m}$, model time step of $0.5\,\mathrm{s}$, and a two-moment cloud microphysics scheme (Seifert and Beheng, 2006) with its default configuration (no enhanced ice crystal growth). The cloud microphysics scheme was extended (Omanovic et al., 2024) by an interface for reading in an external file with the location of the seeding emissions, a prognostic seeding

particle tracer, and a freezing parameterization assuming immersion freezing of the seeding particles (Marcolli et al., 2016; Miller et al., 2025). We emit $10^6$ seeding particles $\mathrm{m^{-3}\,s^{-1}}$ for $6\,\mathrm{min}$ in seven grid boxes, based on the burning time of the seeding flare attached to the UAV and the latitudinal and longitudinal start and end points of the UAV flight track. The seeding plume is identified by ICNC, masked to be $\geq 0.001\,\mathrm{cm^{-3}}$.





**Table 1.** Overview of the simulations discussed in this study showing the name of the simulation, the liquid water path (LWP) at the seeding location ($\mathrm{g\,m^{-2}}$, gives the names for the seeding simulation), the seeding temperature (°C), the time when the seeding was performed (UTC), and the maximum area of the hole identified at the strongest reduction in LWC ($\mathrm{km^2}$, see Fig. 3a).

| Name | LWP ($\mathrm{g\,m^{-2}}$) | Temperature (°C) | Time (UTC) | Max. area of hole ($\mathrm{km^2}$) |
|------|------|------|------|------|
| L25 | 25 | −5.4 | 04:45 | 0.55 |
| L40 | 40 | −5.7 | 04:15 | 0.51 |
| L50 | 50 | −5.6 | 01:00 | 0.78 |
| L60 | 60 | −5.6 | 03:30 | 0.78 |
| L70 | 70 | −5.5 | 03:00 | 0.86 |
| L80 | 80 | −5.5 | 01:20 | 0.84 |
| L100 | 100 | −5.6 | 02:35 | 0.94 |
| L110 | 110 | −5.7 | 02:25 | 1.06 |
| L120 | 120 | −5.7 | 01:45 | 0.81 |

To evaluate the impact of the cloud's LWP on the emergence, size, and lifetime of the hole-punch cloud, we chose a model simulation day, 24 January 2023, that was one of the field experiment days within CLOUDLAB. During this day, we have varying LWPs while temperatures agree with the one from the field experiments ($-5.5 \pm 0.2\,°C$). This way, we ensure that the number of nucleated ice crystals and their subsequent growth remains comparable across the different simulations. However, the model simulation day is a different day to when we observed the hole-punch clouds (12 January 2024), but, as in Omanovic et al. (2024) and Omanovic et al. (2025c), we take a surrogate day. The reasoning is that the model simulations for 12 January 2024 produced a too-short lived and too-shallow cloud to adequately reproduce the observations with no possibility to vary the LWC. Table 1 gives an overview of all simulations, which are named after the LWP at the seeding location in the model. Furthermore, we perturb within the turbulence parameterization (Lilly, 1962; Dipankar et al., 2015) the mixing length, expressed through the Smagorinsky constant $c_s$. Low values of $c_s$ indicate more turbulent flows, while higher values yield more mixing. The default value is set to 0.23 and we chose to perform simulations with a value of 0.17 and 0.3, respectively. The lower value is based on Lilly (1962), who theorized it to be a minimum for turbulent flows. We chose the higher value to be of similar distance to the default value as for 0.17.

## 3 Results

We first show single time steps during the evolution of the seeding plume in L25 (Fig. 2) to give an overview of the seeding simulations. We chose the time steps 1 min, 4 min, and 7 min after the seeding start, such that the first changes in the cloud can be discussed as well as after the seeding concluded (6 min after seeding start). The full evolution of the changes are discussed in the next section. From Fig. 2a−c it is evident that the reductions in LWC are slightly delayed compared to the nucleation of ice crystals, because the ice crystals first need to grow to a certain size to be effective in reducing the LWC (Omanovic





et al., 2024). Nonetheless, the reductions highlight the WBF mechanism. After $1\,\mathrm{min}$ of the seeding start, a fine line appears downstream of the highest ICNC, along which LWC is reduced up to $-100\,\%$, and this signal spreads with time. These stronger

reductions can be visible across a few hundred meters, both in latitude and longitude direction. Note that we chose to show the changes along a constant height, but it is evident that the signal also spreads vertically as shown in Fig. 2d−f. The minimum of the mean reduction at $7\,\mathrm{min}$ coincides with the seeding level as here the highest concentration of ICNC is to be expected. These strong changes in LWC are also visible across time as shown in the last row of Fig. 2h−i, which we discuss in the next section. Here, we first want to assess the impact of varying turbulent mixing lengths acting on the subgrid-scale, expressed by

the Smagorinsky constant $c_s$. A value of $c_s = 0.17$ is identified as a minimum value and indicates more turbulent flows, while a value of 0.3 creates a more homogeneous field by enhancing the mixing. The default value in the model is 0.23 (highlighted in Fig. 2g). The simulated ICNC and absolute and relative changes in LWC are rather insensitive to $c_s$, i.e., their response over time is unchanged. Hence, the subgrid-scale mixing seems not to be a crucial parameter in changing cloud microphysical processes. We hypothesize that the explicitly resolved turbulence (grid-scale) is of higher importance. In the next section, we

computed averages across these three simulations for each seeding simulation. Given the low spread between the perturbations, we are only showing the average and no shading.





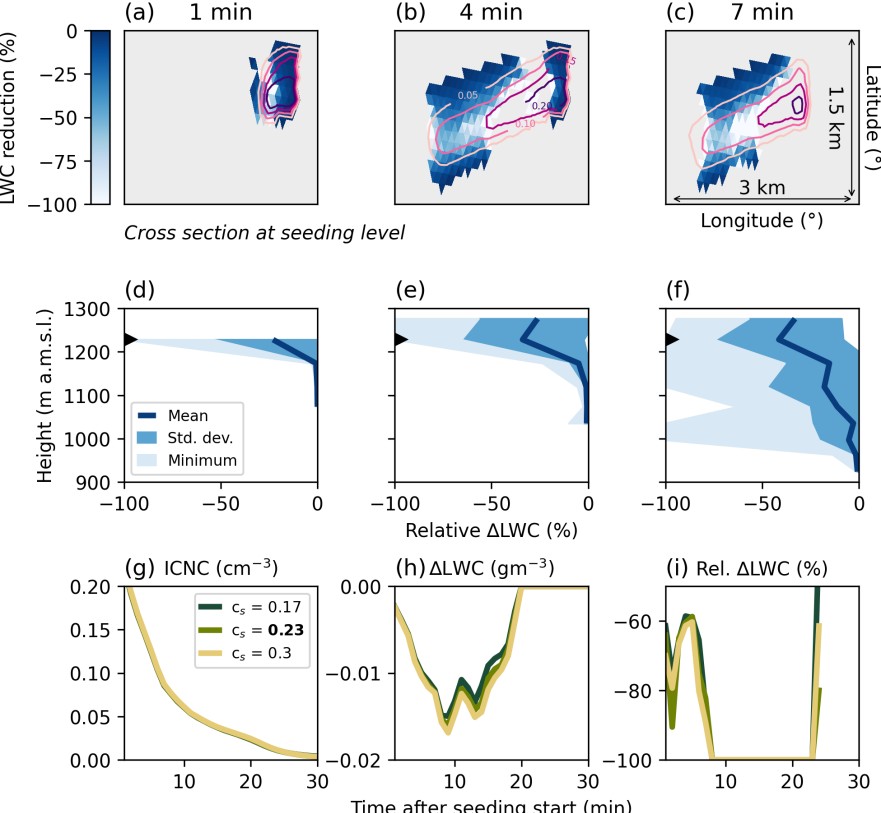

**Figure 2.** Snapshots of seeding simulation L25 with columns indicating the time step after seeding start ($1\,\mathrm{min}$, $4\,\mathrm{min}$, $7\,\mathrm{min}$). The first row (a−c) shows the relative reduction in LWC (%) in blue shading and ICNC ($\mathrm{cm^{-3}}$) in pink contours as a cross section along the seeding level. The domain size is roughly $3 \times 1.5\,\mathrm{km^2}$. The second row (d−f) shows the relative LWC reduction vertically resolved and distilled into mean (dark blue line), standard deviation (medium blue shading), and minimum (light blue shading). The black triangle at $-100\,\%$ / $\approx 1200\,\mathrm{m}$ denotes the seeding level. The third row (g−i) presents the time evolution of the 95[th] percentile in ICNC ($\mathrm{cm^{-3}}$), the 5[th] percentile in absolute changes in LWC ($\mathrm{g\,m^{-3}}$), and the 5[th] percentile relative changes in LWC (%). The different colors represent simulations with varying Smagorinsky constants ($c_s$), where $c_s = 0.23$ is the default value.

Figure 3 summarizes the stronger changes in LWC and cloud optical thickness by computing the 5[th] percentile for each seeding simulation. The cloud optical thickness was calculated as follows: $\tau \approx \int \frac{3}{2} \frac{h(z)\mathrm{LWC}(z)}{\rho_l r_e(z)} dz$, where $h(z)$ refers to the layer thickness in $\mathrm{m}$, $\rho_l$ is the density of liquid water in $\mathrm{kg\,m^{-3}}$, and $r_e$ is the cloud droplet radius in $\mathrm{m}$. In absolute terms, the reductions are stronger for seeding simulations with higher LWPs, because there is more liquid that can be consumed. This is also the case for the timing of the strongest reductions (i.e., minima), which scale in most cases with the initial LWP. We also computed the maximum hole area at these minima and we find a similar pattern: the area of the hole, i.e., strong reductions in LWC, is larger with increasing LWP (see Table 1). At first this seems counterintuitive, however, our hypothesis is that we have two processes defining the hole-punch cloud. Ice crystals form, grow, and lead to the evaporation of cloud droplets through





the WBF mechanism creating the hole. As soon as maximum reductions of $-100\%$ LWC are achieved, no more liquid water is present to be consumed, such that the extent of the hole is limited. To achieve faster and more wide-spread reductions, higher ICNC across a larger area (i.e., longer seeding legs) would be required. Hence, ICNC as well as the surrounding LWC matter for the size of a hole-punch cloud. Furthermore, we identify two clusters: For a low background LWC, we see the first $100\%$ reductions after $7\,\mathrm{min}$ to $8\,\mathrm{min}$, e.g., for L25, L40, and L50. The other simulations form a cluster with minima of LWC

reductions after $11\pm1\,\mathrm{min}$. Hence, a delay in the appearance of strong LWC reductions is notable. It also appears that for seven out of nine seeding simulations a second local minimum is achieved (between 20 and $25\,\mathrm{min}$), which is also associated with $-100\%$ LWC reductions. We hypothesize that as the seeding plume is advected in space, it encounters background states with first lower and then again higher LWCs, such that a renewed reduction in LWC is possible at later times. The cloud optical thickness evolves in accordance with the LWC reductions, where we see an almost complete dissipation of the cloud.

## 4 Conclusions

During the CLOUDLAB field campaigns, we observed the emergence of a hole-punch cloud upon glaciogenic seeding of a low-level stratus cloud. Observations show that strong reductions in LWC coincided with high ICNC, following the WBF mechanism. These findings prompted an investigation into the sensitivity of the seeding-induced hole-punch clouds using numerical simulations. We employed the numerical weather prediction model ICON in large-eddy mode at a horizontal resolution

of $65\,\mathrm{m}$ and simulated one seeding experiment with different starting LWPs, while the temperature and seeding concentration were kept constant. A total of nine seeding simulations were performed with LWPs ranging from $25\,\mathrm{g\,m^{-2}}$ to $120\,\mathrm{g\,m^{-2}}$. For each of the nine simulations, two additional simulations with different turbulent mixing lengths (Smagorinsky constant) were performed.

The first reductions in LWC were associated with the emission of seeding particles at the seeding level, where the first ice

crystals formed. With the dispersion and sedimentation of the ice crystals, reductions in LWC were identified across the entire cloud. Overall, areas of strong LWC reductions were up to $1\,\mathrm{km^2}$. We also found that the turbulent mixing length has little to no impact on the changes in LWC. Focusing on the strong reductions in LWC (absolute and relative) and cloud optical thickness, we identified a delaying effect with increasing LWP: the strongest reductions appeared later in time, with maximum differences of up to $4\,\mathrm{min}$. This behavior was consequently also reflected in the changes in cloud optical thickness, with stronger reductions

$(<-60\%)$ appearing only $10\,\mathrm{min}$ after the seeding start.

This study took on the interesting phenomena of hole-punch clouds, which are known from aircraft flying through a supercooled cloud layer, and investigated them in the context of glaciogenic cloud seeding. Our findings highlighted the interplay of ice crystal growth and the liquid background state of a cloud, i.e., the WBF mechanism. We believe our study provides a foundation for improving our understanding of mixed-phase processes and can inform future cloud seeding strategies.



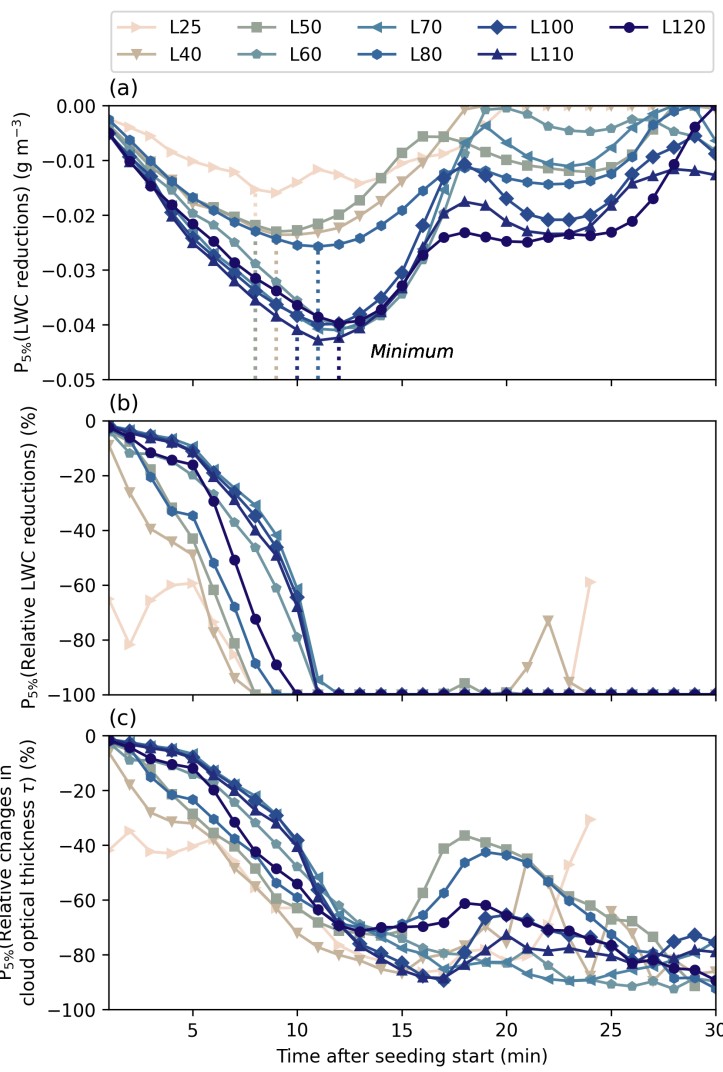

**Figure 3.** For all seeding simulations, differentiated by color and marker, stronger changes indicated by the 5[th] percentile of absolute (a, $g\,m^{-3}$) and relative (b, %) reductions in LWC, and changes in cloud optical thickness (c, %) are shown. The values for each seeding simulation shown are averages over the three simulations with different Smagorinsky constants. In (a) the minima for each seeding simulation are marked by vertical dashed lines, which often overlap as in the case of, e.g., L25, L40, and L50. Note that the minimum of the x-axis corresponds to 1 min after seeding start.



*Code and data availability.* Model and observational data can be found here Omanovic et al. (2025a). Analysis and plotting scripts are available here Omanovic et al. (2025b). We used the ICON model code version 2.6.6 for our simulations. The open-source model code can be obtained at https://icon-model.org/.

*Author contributions.* UL conceived of the idea for CLOUDLAB and obtainted funding. NO and CF were part of the experimental team to design and conduct in-cloud seeding experiments, with conceptual input from UL. CF conducted the analysis of the HOLIMO data. NO set up the model nesting and performed all simulations. DB conducted the first analysis of the simulations in the course of their bachelor thesis. NO refined the analysis and wrote the manuscript. All authors contributed to editing and reviewing the manuscript.

*Competing interests.* The authors declare there are no conflicts of interest for this manuscript.

*Acknowledgements.* The CLOUDLAB project has received funding from the European Research Council (ERC) under the European Union's Horizon 2020 Research and Innovation program (Grant 101021272 CLOUDLAB). This work has been supported by a grant from the Swiss National Supercomputing Centre (CSCS; under project ID lp88). We would like to further extend gratitude to the following people: Jan Henneberger, Anna J. Miller, Fabiola Ramelli, Robert Spirig, and Huiying Zhang for supporting the field experiments, the TROPOS PolarCAP team including Patric Seifert, Kevin Ohneiser, Johannes Bühl, Tom Gaudek, Hannes Griesche, Willi Schimmel, and Martin Radenz for the remote sensing instrumentation and the scientific discussions and collaborations. The Meteomatics drone team, including Lukas Hammerschmidt, Daniel Schmitz, Philipp Kryenbühl, Remo Steiner, and Dominik Brändle for the support, development, and expertise of our drones. Michael Rösch (ETH) for the technical support of our field setup. Maxime Hervo and MeteoSwiss for the wind profiler supporting our experiments. Frank Kasparek and Aleksei Shilin (Cloud Seeding Technologies) for the expertise on our seeding flares. The Swiss Army, and Stefan Minder for allowing the use and maintenance of our main field site.



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
