# Peer review of "Glaciogenic seeding-induced hole-punch clouds and their sensitivity to the clouds' background state"

_EGUsphere, 2025_

## Referee Comment (RC1)

Review for **"Glaciogenic seeding-induced hole-punch clouds and their sensitivity to the clouds' background state"**

**General Comments**

The manuscript by Omanovic et al. explores hole-punch clouds measured during the CLOUDLAB field campaign. Simulations are performed with the NWP model ICON in a large-eddy simulation setup with varying liquid water path and Smagorinsky constants ($c\_s$) to explore the sensitivity of the clouds to these variables. While large impacts are found from a varied LWP, the impact of a varied $c\_s$ is negligible.

This manuscript is nicely written and follows a logical structure. Some parts could benefit from some more clarifications to make the manuscript more approachable for non-experts, please see Specific Comments below. The study is a short but interesting one, providing thus far, unexplored topics, and deserves publication after some revisions. I have no major concerns. Some discussion points are provided below;

**GC1** There is no mention of cloud droplet size distribution. Has this been explored if it has any impact on the rate of glaciation?

**GC2** In regard to L99-L101, have the authors tried to vary the Smagorinsky constant by, say a factor of 10? The values chosen span a small range and it is surprising to see no impacts from varying this constant.

**GC3** In L119 the authors state "We hypothesize that the explicitly resolved turbulence (grid-scale) is of higher importance." To this effect, have the authors explored other horizontal grid spacings? It would be interesting to see how large impact this has on the rate of glaciation.

**Specific Comments**

**L22** "It takes place because of different saturation water vapor pressures over water and ice" While not wrong I would like a longer explanation to make it more accessible to non-experts. Also include the standard statements on the constraint on vapour pressure ($e\_{liq} > e > e\_{ice}$).
**L53** How is the 'void of ice crystals' requirement upstream from the field site ascertained in the experiments?
**L61-62** This sentence is a bit unclear to me. Is the size threshold of 25 mum the limit of detection?
**L63** Please add the reasoning behind the change in analyzing frequency for seeding and non-seeding experiments
**Fig. 1 (d)** Does the y-label indicate that the ceilometer is placed just below 20m? If so, this could perhaps be mentioned in the field site setup to make it clearer.
**L73** Perhaps I'm not aware of the correct terminology but "the attenuated backscatter signal moves higher up" sounds a bit awkward?
**L78** The "we already demonstrated' sounds like it has been done in this manuscript. Perhaps "In previous studies" or similar would be more appropriate.
**L89-95** From a modelling perspective the method makes a lot of sense, but do the authors have an indication whether a hole-punch cloud did occur in the observations during the seeding during this day? And if not, is there an explanation to why that does not occur?
**Fig 2 d- f** Should 'minimum' not rather be 'maximum' as it refers to the largest reduction in LWC? Furthermore, at heights above the seeding level the reductions are very abruply cut off, why is this? Should it not return to 0 rather than no values? Is it constrained by plotting or physics?
In 2 d-f minor xticks could be added to improve readability.

**L118** Perhaps this insensitivity is due to the characteristics of the clouds? This statement seems a

bit too generic for this reason, could the authors add "for these clouds" or similar?

**L129-130** 1) and 2) could be added to better distinguish the processes.

**L132** This would also require a quite homogeneous temperature at seeding height as this is done at quite warm temperatures if I understand the setup correctly. Have the authors looked into the activation rate of the seeding particles in terms of spatial heterogeneity? And how this may impact the obtained ICNC.

**L137-138** As the authors have model output that spans the region of the seeding plume to the field site, could this not be confirmed by evaluating the upstream path?

**Conclusions** I am missing some references and comparisons to other work. As this topic is quite unique I can see that it's hard to find good references, but some references and discussion to whether the reduction in LWP has been previously been seen to this effect should ideally be included here.

One example could be:

"Large-Eddy Simulations of the Impact of Ground-Based Glaciogenic Seeding on Shallow Orographic Convection: A Case Study" Chu et al. 2017

Also a discussion on the varied Smagorinsky constant in comparison to other LES papers would be appropriate, while other papers may not discuss glaciation, a discussion on whether changes in the clouds are seen could be done.

**Technical Corrections**

**L1** "is usually" instead of "usually is"

**L74** add "the" for "the cloud top"